# Structure-Based Identification of Natural Inhibitors Targeting the Gc Glycoprotein of Oropouche Virus: An In Silico Approach

**DOI:** 10.3390/ijms262110541

**Published:** 2025-10-30

**Authors:** Carlos Vargas-Echeverría, Oscar Saurith-Coronell, Juan Rodriguez-Macías, Edgar A. Márquez Brazón, José R. Mora, Fabio Fuentes-Gandara, José L. Paz, Franklin Salazar

**Affiliations:** 1Departamento de Medicina, División Ciencias de la Salud, Universidad del Norte, Km 5, Vía Puerto Colombia, Puerto Colombia 081007, Colombia; vargasce@uninorte.edu.co (C.V.-E.); osaurith@uninorte.edu.co (O.S.-C.); 2Grupo de Investigaciones en Química y Biología, Departamento de Química y Biología, Facultad de Ciencias Básicas, Universidad del Norte, Carrera 51B, Km 5, Vía Puerto Colombia, Barranquilla 081007, Colombia; ebrazon@uninorte.edu.co; 3Facultad de Ciencias de la Salud, Exactas y Naturales, Universidad Libre, Barranquilla 080001, Colombia; 4Grupo de Química Computacional y Teórica (QCT-USFQ), Departamento de Ingeniería Química, Universidad San Francisco de Quito, Diego de Robles y Vía Interoceánica, Quito 170901, Ecuador; jrmora@usfq.edu.ec; 5Department of Natural and Exact Sciences, Universidad de la Costa, Barranquilla 080003, Colombia; ffuentes1@cuc.edu.co; 6Departamento Académico de Química Inorgánica, Facultad de Química e Ingeniería Química, Universidad Nacional Mayor de San Marcos, Lima 15081, Peru; jpazr@unmsm.edu.pe; 7Centro de Química “Dr. Gabriel Chuchani”, Laboratorio de Síntesis Orgánica y Productos Naturales, Instituto Venezolano de Investigaciones Científicas (IVIC), Caracas 1020, Venezuela

**Keywords:** Oropouche virus, plant-derived scaffolds, molecular docking, molecular dynamics simulation, ligand-receptor interactions, ADME-Tox, binding affinity properties

## Abstract

Oropouche virus (OROV), an emerging orthobunyavirus of increasing public health concern in the Americas, currently lacks approved antiviral therapies. In this study, we employed a structure-based in silico approach to identify natural antiviral scaffolds capable of targeting the Gc glycoprotein, a class II fusion protein essential for host membrane fusion and viral entry. A library of 537 plant-derived compounds was screened against the Gc head domain (PDB ID: 6H3X) through molecular docking and redocking, followed by 100-nanosecond molecular dynamics simulations, MM-PBSA free energy calculations, and ADMET profiling. Curcumin and Berberine emerged as standout candidates. Curcumin demonstrated a balanced profile, with stable binding (−38.14 kcal/mol), low backbone RMSD (1.82 Å), and consistent radius of gyration (Rg ~ 18.8 Å), suggesting strong conformational stability and compactness of the protein–ligand complex. Berberine exhibited the most favorable binding energy (−13.10 kcal/mol) and retained dynamic stability (RMSD 1.86 Å; Rg ~ 19.0 Å), though accompanied by predicted cytotoxicity that may require structural refinement. Both compounds induced reduced residue-level fluctuations (RMSF < 2.5 Å) in functionally critical regions of the Gc protein, consistent with a mechanism of action that involves stabilization of the prefusion conformation and interference with the structural transitions required for viral entry. These findings identify curcumin and berberine as promising scaffolds for anti-OROV drug development and offer a rational foundation for future experimental validation targeting viral fusion mechanisms.

## 1. Introduction

Oropouche virus (OROV) is an emerging zoonotic arbovirus that, despite its long history in tropical Latin America, has remained largely overlooked on the global health stage. It belongs to the *Peribunyaviridae* family and *Orthobunyavirus* genus and was first isolated in Trinidad and Tobago in 1955 [1,2,3,4]. Since then, it has been linked to recurrent outbreaks in the Amazon basin and surrounding regions. OROV is transmitted through both sylvatic and urban cycles, primarily involving the biting midge *Culicoides paraensis* and mosquitoes like *Culex quinquefasciatus*, with sloths, non-human primates, and small mammals serving as reservoir hosts [5,6,7].

In recent years, the virus has begun to show signs of expanding beyond its historical ecological niche [2,8,9]. This shift appears to be driven by a combination of environmental disruption, increased human mobility, and the emergence of reassortant viral strains with enhanced fitness. One of the most notable examples was the 2023–2024 outbreak in Brazil, where confirmed cases surged past 10,000, marking a sharp rise compared to the baseline from the previous decade. This event also marked the first time transmission was documented in all 27 federal states [9,10,11,12,13,14,15,16,17,18,19,20,21,22]. Clinically, the outbreak also marked a shift in disease severity, with cases now linked to complications such as Guillain-Barré syndrome, vertical transmission resulting in stillbirth, and potential neurotropism [23,24,25].

What was once considered a localized public health concern now raises alarm on a global scale. Recent reports have confirmed travel-associated cases of OROV infection outside of Latin America, including a case detected in Italy linked to a traveler returning from Cuba [12,17,23,26,27,28]. These findings point to a new and concerning reality: the possibility of international spread into regions unprepared for its diagnosis or containment. As with other arboviruses, the absence of standardized surveillance and diagnostic tools outside endemic areas adds another layer of complexity to outbreak response.

At present, there are no approved antiviral treatments or vaccines against OROV [29,30]. Medical management remains purely supportive, and progress toward targeted therapeutics has been hampered by limited understanding of the virus’s molecular architecture. While structural studies have begun to characterize key viral proteins, including the Gn and Gc glycoproteins encoded by the M segment, efforts to exploit these components pharmacologically remain in their beginning. Among them, the Gc glycoprotein stands out as a promising candidate, as it plays a central role in viral attachment and membrane fusion, making it an attractive molecular target for therapeutic intervention [14,16,19,20].

Recent structural studies have shown that Gc exists in a metastable pre-fusion conformation on the virion surface and undergoes a substantial pH-dependent rearrangement upon endosomal acidification, exposing its fusion loops and promoting host–viral membrane merger [31]. This conformational transition is essential for viral entry and is energetically favorable once triggered; thus, ligands capable of stabilizing the native, pre-fusion form of Gc may effectively prevent the fusion process and block infection. Gc is a class II fusion protein, structurally conserved among several orthobunyaviruses that infect humans, including Bunyamwera virus (BUNV), Germiston virus (GERV), Cache Valley virus (CVV), California encephalitis virus (CEV), Jamestown Canyon virus (JCV), Inkoo virus (INKV), and Ťahyňa virus (TAHV) [32]. Although these viruses are considered neglected and remain understudied at the structural level, experimental evidence has demonstrated that the head domain of the Gc glycoprotein is a key target of neutralizing antibodies [33,34]. This suggests that the Gc head domain plays a critical immunogenic role and highlights its potential as a therapeutic target for small-molecule inhibition strategies.

In the face of growing viral threats with limited therapeutic options, in silico methods have become a powerful and necessary tool in modern antiviral discovery pipelines [35,36]. Their importance is particularly evident when addressing neglected or re-emerging pathogens like OROV, for which experimental models are scarce, and high-containment biosafety facilities limit the feasibility of laboratory research. Computational modeling offers a strategic alternative by simulating molecular interactions between viral targets and potential inhibitors [37,38,39,40,41,42,43,44,45,46,47,48], allowing researchers to generate testable hypotheses with remarkable speed and precision [49,50,51,52]. This approach helps prioritize candidates that not only show high predicted affinity for key viral proteins but also exhibit favorable ADME-Tox profiles. In this context, natural bioactive compounds such as flavonoids, alkaloids, and other phytochemicals stand out for their structural diversity, safety, and well-documented antiviral properties, making them ideal candidates for in silico evaluation [53,54,55,56,57,58,59,60,61].

Beyond their clear advantages in speed and cost, in silico approaches offer an unparalleled window into the molecular mechanics of viral entry, particularly when high-resolution structural data are available, as in the case of the Gc glycoprotein of OROV. By leveraging molecular docking and molecular dynamics simulations, we can visualize in atomic detail how small molecules interact with conserved regions essential for host cell attachment and membrane fusion, revealing structural vulnerabilities that may be exploited therapeutically.

To the best of our knowledge, this is the first study to comprehensively evaluate the interaction of plant-derived scaffolds with the Gc glycoprotein of OROV using a multiparametric computational strategy. By targeting a conserved region within the head domain, we identified promising candidates with favorable binding profiles and drug-like properties (as shown in Figure 1). These findings contribute to a growing conceptual framework for inhibiting OROV entry and support the rational design of antiviral agents against this neglected yet increasingly relevant pathogen.

## 2. Results and Discussion

### 2.1. Data Collection

#### 2.1.1. Selection of Luteolin as Baseline Scaffold

Luteolin was selected as the baseline scaffold in our computational screening due to its well-established bioactive profile, particularly its potent antiviral, anti-inflammatory, and antioxidant properties. As a naturally occurring flavonoid found in a variety of fruits, vegetables, and medicinal plants, luteolin has demonstrated the ability to modulate key cellular pathways involved in viral infection and replication, including MAPK, PI3K-AKT, and NF-κB signaling [62]. These mechanisms are frequently hijacked by viruses to facilitate their life cycle, making luteolin a compelling candidate for antiviral drug discovery.

Recent studies have shown that luteolin exhibits broad-spectrum antiviral activity against several human and animal viruses, including coronaviruses, influenza virus, herpes simplex virus, and respiratory syncytial virus. Its antiviral effects are mediated through multiple mechanisms, such as inhibition of viral entry, suppression of viral replication, and enhancement of host immune responses [63,64,65,66]. From a structural perspective, luteolin’s polyphenolic backbone and moderate lipophilicity make it a favorable scaffold for computational docking studies. Its ability to form hydrogen bonds and engage in π–π interactions allows it to interact effectively with hydrophobic and polar residues within viral protein targets [67]. Additionally, efforts to improve its bioavailability, such as the development of luteolin–phospholipid complexes, have shown promise in enhancing its pharmacokinetic properties without compromising its biological activity. Therefore, luteolin analog searching was conducted using the pubchem web page algorithm for similarity [68].

Given this profile, luteolin was used as the reference scaffold for similarity-based virtual screening, and analogues were retrieved using the PubChem similarity search tool [68]. However, for docking and MD simulations, quercetin was selected as the positive control based on prior experimental reports demonstrating its antiviral activity specifically against OROV [54]. This distinction allowed us to use luteolin for rational library design while benchmarking docking predictions against a compound with validated biological activity.

#### 2.1.2. Similarity-Based Compound Retrieval Using PubChem: 2D and 3D Approaches

From an initial pool of 270 flavonoids and 267 alkaloids, 20 natural compounds were selected based on their reported antiviral activity, structural diversity, and pharmacological promise. This process incorporated both 2D and 3D similarity searches using the PubChem platform (https://pubchem.ncbi.nlm.nih.gov, accessed on 20 May 2025), applying the *“Drug and Medication”* filter to prioritize compounds with known or potential bioactivity. The selected compounds (Figure 2) were retrieved in SMILES and SDF formats for subsequent docking and molecular dynamics simulations.

### 2.2. Molecular Docking

#### Target Selection

In selecting the Gc glycoprotein of OROV as the focus for our docking studies, we were guided by both structural and functional evidence. Gc, a class II fusion protein, forms the apex of the trimeric viral spike and mediates the crucial membrane fusion step during host cell interaction [69]. High-resolution structures of the OROV Gc head domain (e.g., PDB: 6H3X) provide an accurate template for in silico analysis [69], allowing us to explore ligand–protein interactions with confidence. Notably, the N-terminal “head” of Gc is both surface-exposed and antigenically significant [69], making it a prime candidate for small-molecule binding that could potentially obstruct conformational changes required for fusion. Recent structural studies have shown that Gc exists in a metastable pre-fusion conformation on the virion surface and undergoes a substantial pH-dependent rearrangement upon endosomal acidification, exposing its fusion loops and promoting host–viral membrane merger [31]. This conformational transition is essential for viral entry and is energetically favorable once triggered; thus, ligands capable of stabilizing the native, pre-fusion form of Gc may effectively prevent the fusion process and block infection. The Gn and Gc proteins, produced from the virus’s M segment, work together to help the virus attach to host cells; therefore, it is a crucial first step in infection. By targeting the Gc protein specifically, we may be able to interrupt this early stage and prevent the virus from starting its replication process. Moreover, interest in Gc as an antiviral target is growing; recent computational work identified strong binding of natural flavonoids to the Gc [48]. On the other hand, Menezes et al. [54] successfully docked quercetin hydrate to the Gc protein, suggesting a new antiviral. These studies validate the suitability of Gc for structure-based drug discovery and support our strategic focus. In this line, this work uses the structure of OROV Gc glycoprotein head domain, retrieved from the RCSB Protein Data Bank (PDB ID: 6H3X). The quaternary structure of this protein domain, after it is cleaned, is depicted in Figure 3.

### 2.3. Docking Results

Using 6H3X as a target, and the compounds shown in Figure 2 as ligands, the docking results are summarized in Table 1, while the 2D diagrams for the compounds with the highest scoring values are shown in Figure 4. This table shows not only the scoring values for docking but also the differences in molecular interactions (hydrogen bond and other Van der Waals or hydrophobic interactions).

It is important to note that even though the luteolin was used as a base scaffold for the analogues searching, we decided to use the quercetin as a positive control due to previously reported experimental activity against OROV, serving as a validated comparator to evaluate the potential of the screened compounds [54]. It means that scoring values higher than quercetin were considered “actives”.

According to Table 1, curcumin exhibited the most favorable binding energy at −7.9 kcal/mol, surpassing quercetin and suggesting a strong potential for stable interaction with the viral target. Its docking pose revealed multiple hydrogen bonds with residues such as Tyr570, Arg604, Ala605, Ser607, and Asn608, as well as hydrophobic contacts involving Ile606 and Asp609. This interaction pattern indicates a stable binding orientation within the proposed pocket of the Gc glycoprotein, potentially contributing to modulation of its structural or functional dynamics.

Baicalin and naringin also demonstrated promising binding energies of −7.4 and −7.3 kcal/mol, respectively. Baicalin formed hydrogen bonds with the amino acid residues Ala605, Met611, and Leu676, indicating specific and stable interactions. In comparison, naringin interacted with a broader set of residues, including Tyr570, Ala605, and Asp609, which suggests a more extensive and potentially more favorable positioning within the binding pocket.

Compounds such as biochanin a, luteolin, and cynaroside also demonstrated strong binding affinities, with energies ranging from −7.2 to −6.9 kcal/mol. These molecules consistently interacted with key residues like Ala605 and Tyr675, which appear to serve as common anchoring sites for multiple ligands within the binding pocket.

In contrast, compounds with binding energies below −6.5 kcal/mol, such as fisetin and apigenin, while still interacting with key residues, may exhibit comparatively weaker binding stability under physiological conditions.

These results highlight curcumin, baicalin, and naringin as the most promising candidates, with binding affinities that are comparable to or slightly stronger than that of quercetin. Their interaction profiles suggest a meaningful potential to disrupt the conformational dynamics of the Gc glycoprotein, encouraging further exploration through molecular dynamics simulations and, ideally, experimental validation. Quercetin was selected as a reference compound based on its previously reported antiviral effects against OROV and its performance in molecular docking studies. In particular, the study by Menezes et al. explored quercetin hydrate as a potential antiviral agent against OROV and demonstrated favorable interaction with the viral target [54], supporting its use as a positive control in our computational pipeline.

### 2.4. Pharmacokinetic and Toxicity Properties

To comprehensively assess the therapeutic potential of the selected plant-derived scaffolds, namely curcumin, baicalin, naringin, lonicerin, cynaroside, berberine, and quercetin (as a positive control), we performed an in silico ADMET profiling using SwissADME and ProTox v3.0 [43,47,70,71], with key findings summarized in Table 2. The six compounds were prioritized for this analysis based on their favorable docking free energies (≤−7.0 kcal/mol), a commonly accepted threshold for strong binding affinity in molecular docking studies, and their consistent interaction profiles with the Gc glycoprotein in virtual screening [72]. This analysis was complemented by the evaluation of physicochemical properties and synthetic accessibility to construct a multidimensional profile of each compound′s drug-likeness. Quercetin, a well-characterized flavonol with prior evidence of antiviral activity against arboviruses, was included as a positive control [54]. Its favorable profile, including high gastrointestinal (GI) absorption, no violations of Lipinski’s rules, moderate lipophilicity (Log P = 1.23), and low predicted toxicity, validates its use as a benchmark for assessing novel antiviral candidates.

Among the test scaffolds, curcumin and berberine emerged as the most promising from a pharmacokinetic standpoint. Both exhibited high predicted GI absorption and conformed to Lipinski’s rule of five [73], indicating good oral bioavailability. Curcumin’s physicochemical attributes, including a moderate molecular weight (368.38 g/mol), low rotatable bond count, and balanced hydrogen-bonding capacity, likely contribute to its favorable absorption and permeability. Furthermore, its low synthetic accessibility score (2.97) supports its viability as a lead compound for derivatization or formulation. Berberine, while exhibiting the highest lipophilicity (Log P = 2.53) and being the only compound predicted to cross the blood–brain barrier, also displayed potential red flags in its safety profile. In silico toxicity models flagged it as potentially cytotoxic, mutagenic, and neurotoxic, with an estimated LD_50_ of 200 mg/kg, placing it in toxicity class 3, which corresponds to “toxic if swallowed” (LD_50_ between 50 and 300 mg/kg according to the ProTox-III classification). This suggests a narrower therapeutic window and the need for structural optimization to mitigate adverse effects.

In contrast, the glycosylated flavonoids (baicalin, naringin, lonicerin, and cynaroside) demonstrated poor predicted GI absorption, likely due to their higher molecular weights (greater than 440 g/mol), elevated numbers of hydrogen bond donors and acceptors, and repeated violations of Lipinski’s criteria. Their complex structures also pose synthetic challenges, as reflected in synthetic accessibility scores exceeding 5.0. Nevertheless, these compounds exhibited consistently favorable toxicity profiles, with no predicted hepatotoxic or mutagenic effects and high LD_50_ values above 5000 mg/kg, placing them in toxicity class 5 or 6. These classes are defined as “may be harmful” (class 5, LD_50_ between 2000 and 5000 mg/kg) or “non-toxic” (class 6, LD_50_ greater than 5000 mg/kg), respectively. Their moderate aqueous solubility (Log S between −3.4 and −3.6) suggests that non-oral delivery strategies, such as pulmonary or localized administration, may be more appropriate to circumvent systemic bioavailability limitations.

Overall, these ADMET-based insights reveal the delicate interplay between pharmacokinetics, safety, and drug-likeness that defines the therapeutic potential of each compound. Curcumin emerges as a particularly promising candidate, showing good absorption, low predicted toxicity, and ease of chemical synthesis. Although berberine raises some safety concerns, its strong permeability and potential to cross into the central nervous system suggest it still holds therapeutic value. Meanwhile, the glycosylated flavonoids, despite their lower permeability, present unique, non-toxic molecular frameworks that could be further optimized through targeted or alternative delivery methods. These findings provide important context for interpreting the molecular docking and dynamics results and support the selection of quercetin as a reliable reference compound in the ongoing search for new anti-OROV agents.

### 2.5. Molecular Dynamics Simulation

We simulated the structures of all selected biomolecules–protein complexes in an aqueous environment to evaluate their dynamic behavior and stability against the Gc glycoprotein head domain of the OROV. For the molecular dynamics simulations, compound selection was guided by a multifactorial rationale that extended beyond docking scores alone. We prioritized those candidates that not only demonstrated strong binding affinities and stable interactions with conserved residues within the Gc head domain (PDB ID: 6H3X) but also presented a favorable pharmacokinetic and safety profile. Emphasis was placed on molecules with low predicted systemic toxicity and acceptable synthetic accessibility, as these features collectively increase the likelihood of successful translational development. This integrative selection strategy ensured that the compounds advancing to dynamic simulations were not only potent in silico binders but also carried realistic therapeutic potential. Moreover, some amino acids are present in the molecular interaction with the compounds with the most scoring values, such as Tyr635, Met643, Asp665, and Leu702, which are positioned within the binding pocket of the Gc glycoprotein and have been associated with ligand engagement in previous structural analyses (as shown in Table 1).

To evaluate the conformational stability of the predicted ligand–protein complexes under near-physiological conditions, we performed all-atom molecular dynamics (MD) simulations over a 100-nanosecond timescale and monitored root-mean-square deviation (RMSD) as the primary metric of structural fluctuation, as shown in Figure 5. RMSD is a widely recognized parameter for assessing how much a protein structure shifts over time relative to its initial configuration, particularly at the backbone level, and it serves as a practical proxy for gauging the stability of ligand binding and the integrity of the protein–ligand interface [47].

All simulated systems followed a consistent dynamic pattern, with an initial phase of adjustment during the first 5 nanoseconds. This early fluctuation is expected, as the receptor accommodates the ligand into its binding cavity, often involving subtle rearrangements of side chains or solvent-exposed loops. After this equilibration period, all complexes reached a relatively stable plateau, maintaining low-amplitude RMSD values throughout the remainder of the simulation. Mean RMSD values ranged from 1.78 Å to 1.91 Å as shown in Table 3, remaining well within the range typically associated with stable protein–ligand interactions.

Among the compounds tested, naringin (1.78 Å) and baicalin (1.79 Å) showed the lowest mean RMSD values, suggesting not only rapid convergence but also consistent spatial retention within the binding site. These profiles point to well-accommodated binding modes with minimal disruption to the protein architecture. Curcumin, cynaroside, and berberine followed closely, with average RMSD values of 1.82 Å, 1.82 Å, and 1.86 Å, respectively. These differences, although modest, may reflect compound-specific flexibility or subtle variations in local binding pocket topology.

Of particular note, cynaroside, berberine, and quercetin exhibited strikingly stable behavior during the second half of the simulation window (50 to 100 ns). Their respective standard deviations were 0.14 Å for cynaroside, 0.17 Å for berberine, and 0.13 Å for quercetin, among the lowest observed, indicating minimal fluctuation around their equilibrium positions once bound. This long-term stability may be reflective of a deeply embedded binding mode, supported by favorable interactions with key conserved residues identified in our docking analyses. Such persistence is relevant from a pharmacodynamic perspective, as it suggests potential for sustained inhibition during the viral entry process, where Gc-mediated fusion is transient yet critical.

By contrast, lonicerin, though still displaying acceptable overall stability (mean RMSD = 1.87 Å), showed the largest deviation across the simulation (σ = 0.22 Å), consistent with a more flexible binding profile. While this may suggest a less rigid anchoring within the receptor pocket, it could also reflect adaptability in interacting with dynamic or partially disordered regions of the viral fusion domain. These properties are increasingly recognized as valuable in drug discovery, particularly when targeting highly variable or conformationally labile viral proteins.

For viral fusion proteins like the Gc glycoprotein, stability of the ligand–protein complex is crucial. These proteins undergo conformational rearrangements to mediate membrane fusion. A compound that locks the protein in a pre-fusion state, as suggested by low RMSD, could effectively inhibit viral entry. Therefore, compounds with consistently low RMSD values (e.g., cynaroside, quercetin, curcumin) may act by stabilizing the glycoprotein in an inactive conformation, preventing the structural transitions necessary for infection; therefore, even though RMSD values are not just technical metrics, they can be mechanistically informative. In this study, compounds with low and stable RMSD profiles are likely to be mechanistically aligned with fusion inhibition, making them strong candidates for further validation.

To complement the conformational and dynamic characterization of the ligand–protein interactions observed during molecular dynamics (MD) simulations, we carried out an energetic analysis using the MM-PBSA (Molecular Mechanics Poisson–Boltzmann Surface Area) methodology. This well-established approach allows for the estimation of the binding free energy of biomolecular complexes under near-physiological conditions, providing a thermodynamic perspective on ligand affinity and stability.

The Molecular Mechanics Poisson–Boltzmann Surface Area (MM-PBSA) method is widely recognized for its robustness in estimating binding free energies between ligands and target proteins within biomolecular systems. In this study, we employed MM-PBSA calculations based on molecular dynamics (MD) simulations performed in YASARA Structure to quantify the interaction energies of the selected compounds. By default, YASARA reports binding affinity as positive values, where higher values denote stronger interactions [43,71,72]. However, negative values produced by the standard macro do not necessarily imply the absence of binding, but rather comparatively weaker association.

To provide a comparative setpoint, we included quercetin as a positive control. Although it has not previously been studied in the context of the Gc head domain, its documented antiviral activity and binding capacity to other structural proteins of the OROV make it a suitable reference molecule. In our simulations, quercetin demonstrated a binding energy of −39.79 kcal/mol, consistent with a moderate-to-strong interaction and validating its role as a standard for comparison.

Relative to quercetin, berberine showed the most favorable binding profile, with a binding energy of −13.10 kcal/mol, indicating the strongest predicted affinity among all compounds tested. This result was further supported by the lowest standard deviation (σ = 8.89 kcal/mol), suggesting that the interaction remained consistent and stable throughout the 100 ns simulation. Curcumin (−38.14 kcal/mol, σ = 14.25 kcal/mol) also exhibited a binding energy comparable to quercetin, suggesting similarly favorable engagement with the Gc head domain.

Other compounds such as cynaroside (−50.81 kcal/mol), naringin (−58.01 kcal/mol), and baicalin (−62.84 kcal/mol) displayed more negative binding energies, reflecting moderately weaker thermodynamic profiles. These compounds also exhibited higher fluctuations (σ = 17.49, 15.75, and 15.61 kcal/mol, respectively), which may indicate more flexible binding or transient interactions. Nonetheless, their energies remain within a range compatible with specific binding and could be of interest for further optimization.

Finally, lonicerin presented the most negative binding energy (−92.91 kcal/mol), placing it at the lower end of the affinity scale in this dataset. While this may suggest a less favorable interaction energetically, it does not necessarily preclude biological activity, especially when considered in combination with other structural and dynamic parameters, such as conformational retention observed in RMSD analyses.

According to these results, the MM-PBSA-based binding energy analysis identifies berberine, curcumin, and quercetin as the most promising ligands based on both thermodynamic favorability and interaction stability, as shown in Figure 6. These energy patterns are indicative of ligand-induced stabilization of the Gc glycoprotein, potentially locking it in a pre-fusion conformation and thereby inhibiting the conformational changes required for membrane fusion and viral entry. This proposed mechanism aligns with findings from other class II fusion glycoproteins, where small molecules have been shown to stabilize metastable prefusion states, effectively preventing the structural rearrangements necessary for viral–host membrane fusion [33,74]. Such a mechanism has been described in studies targeting flavivirus and bunyavirus envelope proteins, where binding energy stability during simulation correlates with functional inhibition of viral entry [31].

#### 2.5.1. Residue-Level Flexibility (RMSF) and Mechanistic Implications

The RMSF analysis provides a residue-by-residue view of the flexibility of the Gc glycoprotein upon ligand binding. In this study, the RMSF profiles for all ligand–protein complexes, including curcumin, quercetin, cynaroside, and others, reveal localized fluctuations, with most residues exhibiting low to moderate mobility (generally <2.5 Å), suggesting that ligand binding contributes to structural stabilization of the protein, as shown in Figure 7.

Notably, compounds such as curcumin, quercetin, and cynaroside induce lower RMSF values across key functional regions of the Gc glycoprotein, particularly in domains associated with membrane fusion. This reduced flexibility supports the hypothesis that these ligands may act by restricting the conformational dynamics required for the glycoprotein to transition from its pre-fusion to post-fusion state, a critical step in viral entry. This interpretation is consistent with previous studies on class II viral fusion proteins, where ligand-induced rigidity in fusion loops or hinge regions has been associated with inhibition of membrane fusion and viral infectivity [75,76]. Therefore, these results represent strong evidence supporting these compounds may function as conformational stabilizers, locking the Gc glycoprotein in an inactive state and preventing the structural rearrangements necessary for host cell entry.

#### 2.5.2. Radius of Gyration Analysis and Structural Implications

The radius of gyration (Rg) is a critical parameter in molecular dynamics that reflects the overall compactness and structural stability of a protein or protein–ligand complex over time. In this study, the Rg profiles of the Gc glycoprotein complex, jointly with the selected compounds, were monitored over a 100 ns simulation.

The results reveal that most complexes maintained Rg values within a narrow range (~18.6–19.4 Å), indicating that the global fold of the Gc glycoprotein remained stable throughout the simulation, as shown in Figure 8. Notably, compounds such as curcumin and quercetin exhibited consistently low and stable Rg values, suggesting that their binding contributes to a more compact and conformationally restrained protein structure. This observation aligns with the proposed mechanism of action, wherein these ligands may act by stabilizing the prefusion conformation of the Gc glycoprotein, thereby hindering the structural rearrangements necessary for membrane fusion and viral entry.

In contrast, minor fluctuations observed with compounds like berberine and lonicerin may reflect transient conformational or less rigid binding, which could influence their inhibitory potential. These findings are consistent with previous studies demonstrating that ligand-induced reductions in Rg often correlate with enhanced structural integrity and functional inhibition of viral fusion proteins. Thus, the Rg analysis not only supports the mechanistic hypothesis proposed from RMSD and binding energy data but also reinforces the therapeutic promise of the most stable ligands.

An integrative analysis combining molecular dynamics, MM-PBSA energy calculations, and ADMET profiling provided a comprehensive view of how selected flavonoid and alkaloid scaffolds engage the Gc glycoprotein of OROV. Among the 20 plant-derived compounds evaluated, berberine and curcumin consistently emerged as the most promising candidates, albeit through distinct but complementary mechanisms of action.

Berberine exhibited the strongest predicted binding affinity in the series, with a binding free energy average of −13.10 kcal/mol, indicating a highly favorable thermodynamic interaction with the target. This was further supported by its stable dynamic behavior throughout the simulation, as reflected in low root-mean-square deviation (RMSD) values and a consistently compact radius of gyration (Figure 8), suggesting that the complex maintained its integrity over time. RMSF analysis (Figure 7) revealed low atomic fluctuations in the ligand-binding interface, reinforcing the idea of a stable and tightly held interaction. Structurally, Berberine’s rigid, planar isoquinoline scaffold with a permanent positive charge facilitates electrostatic and aromatic stacking interactions with key residues inside the pocket. The compound has only two rotatable bonds, which contribute to its limited conformational drift during the simulation.

In terms of hydrogen bonding, Berberine maintained a consistent profile (Figure 8), further supporting its structural compatibility with the Gc binding site. Its pharmacokinetic properties were also encouraging, showing high predicted gastrointestinal absorption and suitable lipophilicity. However, toxicity predictions, including a high probability of cytotoxicity and mutagenicity, along with a low LD_50_ estimate of 200 mg/kg, highlight the need for further optimization to improve its safety profile. Despite these limitations, berberine was included among the evaluated candidates due to its favorable docking performance and its broad-spectrum antiviral activity previously reported against viruses such as SARS-CoV-2, dengue, influenza, enterovirus, and herpesvirus [60,77,78,79,80,81]. These data support its pharmacological relevance; however, we acknowledge that its toxicity must be carefully addressed, and any future consideration would require rational structural optimization and toxicity mitigation strategies.

In contrast, curcumin displayed a more moderate binding affinity of −38.14 kcal/mol but showed outstanding pharmacokinetic and dynamic performance. Its symmetric scaffold, composed of aromatic rings connected by a conjugated β-diketone bridge, allows for flexible yet stable interactions through hydrogen bonding and π–π stacking [75,76,82]. This structural adaptability translated into excellent dynamic stability, with low RMSD and minimal fluctuations in the simulation, as seen in the RMSF data (Figure 7). Its radius of gyration remained consistent with those observed in stable ligand–protein complexes (Figure 6), sustained interactions with the Gc head domain. Curcumin also exhibited a clean ADMET profile, with no major toxicity flags, high gastrointestinal absorption, full compliance with Lipinski′s rules, and favorable synthetic accessibility. Together, these attributes make curcumin a strong candidate for further development as a Gc-targeting antiviral.

Quercetin, used here as a positive control due to its established antiviral profile, demonstrated dynamic and energetic behavior comparable to curcumin. Its performance validated the predictive model and reinforced the relevance of curcumin and berberine as equal or potentially superior candidates.

Other compounds, such as cynaroside, lonicerin, baicalin, and naringin, displayed adequate structural stability during the simulations but were limited by suboptimal ADMET characteristics. These included large molecular weights, excessive hydrogen bonding capacity, and low lipophilicity, all of which may compromise oral bioavailability and synthetic tractability.

Overall, the findings suggest that both berberine and curcumin hold meaningful potential to disrupt OROV infection by engaging conformationally sensitive regions of the Gc glycoprotein. Berberine, owing to its rigid and positively charged scaffold, appears particularly well-suited to stabilize the protein in a non-functional conformation. This may prevent the structural rearrangements required for membrane fusion (as shown in Figure 9). In contrast, curcumin exhibits a more flexible and adaptive binding profile, which could interfere earlier in the viral entry process, possibly by modulating the interaction between the viral envelope and the host membrane. Together, these two natural compounds represent complementary antiviral strategies: one focused on tight structural stabilization of the target, and the other on adaptable, biocompatible engagement. Their distinct yet synergistic profiles provide a compelling rationale for further development of plant-derived scaffolds as therapeutic candidates against OROV.

## 3. Materials and Methods

### 3.1. Compound Selection and Data Collection

#### Similarity-Based Compound Retrieval Using PubChem: 2D and 3D Approaches

To expand our library of potential antiviral candidates, we utilized the PubChem similarity search platform to identify compounds structurally related to luteolin. Pubchem offers two complementary approaches for molecular similarity assessment: 2D fingerprint-based and 3D conformer-based searches, each with distinct advantages in virtual screening workflows. In this study, we primarily employed the 2D similarity search to rapidly generate a focused library of luteolin analogs. However, we also considered 3D conformer data where available to ensure that selected candidates retained favorable spatial characteristics for potential interaction with the Gc glycoprotein of the OROV.

The 2D similarity search works by comparing simplified digital representations of molecules, known as molecular fingerprints. These fingerprints are essentially strings of binary digits that indicate whether certain structural features are present in a molecule. To measure how similar two molecules are, the method uses the Tanimoto coefficient, a commonly used metric that calculates the degree of shared features between them. This approach is both fast and effective, especially when looking for compounds that have similar core structures or functional groups to a reference molecule.

Unlike 2D methods, 3D similarity searches assess the spatial shape and electrostatic properties of molecules. They work by comparing conformers, which are three-dimensional arrangements of atoms that often represent the biologically active form of a compound. Using shape-based alignment algorithms, this approach can identify molecules with similar pharmacophoric features even when their 2D structures differ significantly. Although it is more computationally demanding, 3D similarity is particularly valuable when biological activity is closely linked to molecular conformation, a common scenario in drug discovery [79].

### 3.2. Ligand Preparation and Optimization

All selected molecules were first energy-minimized using Avogadro (v1.2) with the MMFF94 force field to ensure stable and optimized conformations. Further refinements and topological checks were conducted using Chem3D^®^ Professional 17.1 (PerkinElmer, Waltham, MA, USA). Structures were then exported in the appropriate formats required for docking analysis, with special attention given to ensuring consistency in protonation states and geometry.

### 3.3. ADMET Predictions

To assess the drug-likeness and safety profiles of the compounds, we evaluated their pharmacokinetic and toxicity parameters using SwissADME (http://www.swissadme.ch/, accessed on 27 May 2025) and ProTox 3.0 (https://tox-new.charite.de/protox_III/, accessed on 27 May 2025). These platforms provided valuable insight into parameters such as oral bioavailability, gastrointestinal absorption, blood–brain barrier permeability, CYP450 enzyme inhibition, and potential toxicity classes. Input data were submitted in either SMILES or SDF format, and default settings were used for all analyses. Compounds with unfavorable pharmacokinetics or high predicted toxicity were deprioritized before proceeding to docking simulations.

### 3.4. Molecular Docking Studies

To evaluate how well the compounds bind to the viral Gc protein, we used the CB-Dock2 server (https://cadd.labshare.cn/cb-dock2, accessed on 20 January 2025) [69,77,83,84,85], which offers blind docking powered by AutoDock Vina. The protein structure (6H3X) was first cleaned using PyMOL 2.1 by removing crystallographic water molecules, ions, and co-crystallized ligands. Polar hydrogens were added, and Kollman charges were assigned as needed.

Docking was performed for each compound individually, and binding affinities were recorded in kcal/mol. We also analyzed the binding pockets and identified the key residues involved in hydrogen bonding and hydrophobic interactions. These data were used to rank the ligands based on their predicted binding strength and interaction profile, giving us a clearer picture of their potential inhibitory effects on the Gc protein.

### 3.5. Molecular Dynamics Simulations

To explore how stable these interactions would be under dynamic, physiological conditions, we selected the top six ligands based on docking scores and interaction profiles: curcumin, baicalin, naringin, cynaroside, berberine, and quercetin (as positive control). These compounds were subjected to molecular dynamics (MD) simulations using YASARA Dynamics [54,73,74,86,87,88], with the AMBER14 force field. 

Each protein–ligand complex was placed in a cubic simulation box filled with explicit TIP3P water molecules and surrounded by a 3.5 Å solvent buffer. The systems were neutralized with 0.9% NaCl to mimic physiological conditions. Simulations were carried out under standard temperature (298 K) and pH (7.4), using periodic boundary conditions. This adjustment is achieved through a built-in chemical knowledge base that assigns pKa values to functional groups using SMILES strings. Based on these values, YASARA determines the most appropriate protonation pattern for each residue at the chosen pH, in this case, pH 7.4. This ensures that the system reflects the correct protonation state distribution under physiological conditions. An initial equilibration step was carried out using the NVT ensemble (constant number of particles, volume, and temperature), with approximately 29,900 water molecules and a temperature of 293 K. The box volume varied depending on the size of each complex. Following equilibration, production simulations were conducted using the NPT ensemble (constant number of particles, pressure, and temperature), with pressure set to 1 atm and temperature maintained at 293 K. Temperature and pressure were regulated using the Berendsen thermostat and barostat, with time constants of 0.1 ps and 2.0 ps, respectively. This setup ensures smooth equilibration and realistic solvent behavior, allowing the system volume to adjust naturally during the simulation.

After energy minimization, 100-nanosecond simulations were run for each complex. The simulation snapshots were saved every 100 picoseconds for analysis. We used the built-in YASARA macros (md_analyze_dynamics.mcr) and (md_analyzebindenergy.mcr) to compute structural and energetic parameters such as RMSD, radius of gyration, hydrogen bond persistence, and binding free energy (using the adaptive Poisson–Boltzmann solver, APBS). These simulations allowed us to assess not just how well the ligands bound initially, but how stable those interactions remained over time, critical insight for evaluating real-world drug potential.

## 4. Conclusions

This study presents a comprehensive computational investigation into the antiviral potential of plant-derived compounds targeting the Gc glycoprotein of OROV, a critical mediator of viral membrane fusion and a compelling molecular target for therapeutic intervention. By integrating molecular docking, dynamics simulations, free energy calculations, and ADMET profiling, we identified curcumin and berberine as two promising scaffolds with complementary pharmacological profiles. Curcumin demonstrated consistent structural stability throughout the 100-nanosecond simulation, reflected by low RMSD and Rg values, stable hydrogen bonding, and minimal residue-level fluctuations. It has favorable pharmacokinetic properties, high predicted gastrointestinal absorption, low toxicity risk, and synthetic tractability. Berberine, on the other hand, exhibited the strongest binding affinity among all candidates and formed a compact, stable complex during molecular dynamics simulations. However, we acknowledge that its toxicity profile warrants caution. Rather than promoting berberine as a direct therapeutic candidate, we propose it as a potential lead scaffold for future structure-based optimization. Its rigid polycyclic framework and favorable interaction dynamics make it an attractive template for the design of safer derivatives. Moreover, RMSF and Rg analyses suggest that its inhibitory effect may involve stabilizing the Gc glycoprotein in its prefusion conformation, potentially interfering with the structural transitions required for membrane fusion. This conformational locking could prevent the structural rearrangements necessary for viral entry, a mechanism consistent with previous reports on class II fusion inhibitors in other arboviruses.

To support these findings, future in vitro confirmatory experiments could include pseudotyped virus entry assays, cell–cell fusion assays, or plaque reduction neutralization tests using Gc-expressing systems. These approaches would provide functional validation of the predicted inhibitory effects and clarify the compounds’ ability to interfere with OROV entry. While further in vitro and in vivo studies are needed, the evidence presented here lays a solid foundation for the continued development of curcumin-, berberine-, and flavonoid-based scaffolds as antiviral agents against OROV and potentially other emerging orthobunyaviruses.

## Figures and Tables

**Figure 1 ijms-26-10541-f001:**
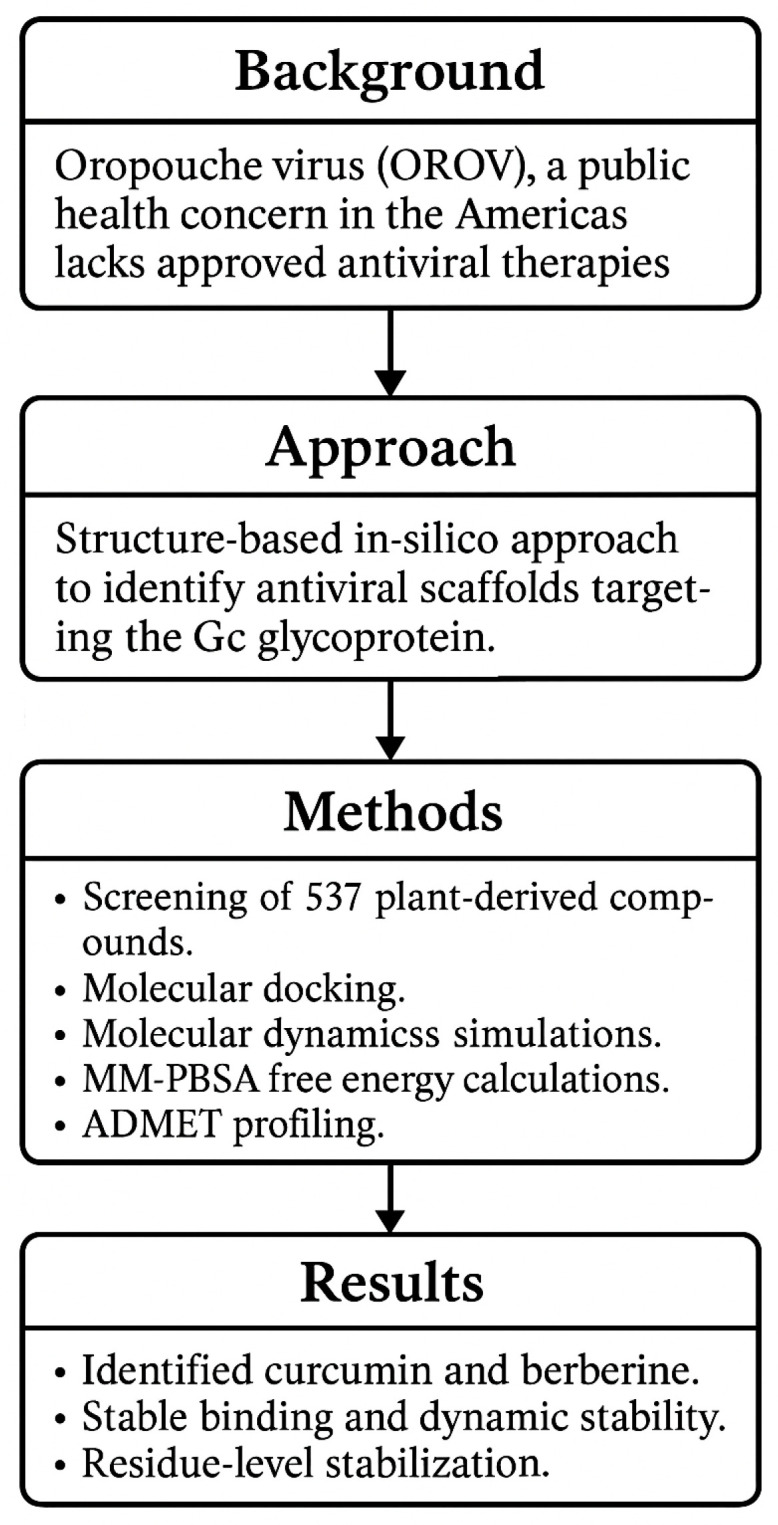
Workflow summarizing the in silico identification of natural inhibitors targeting the Gc glycoprotein of OROV.

**Figure 2 ijms-26-10541-f002:**
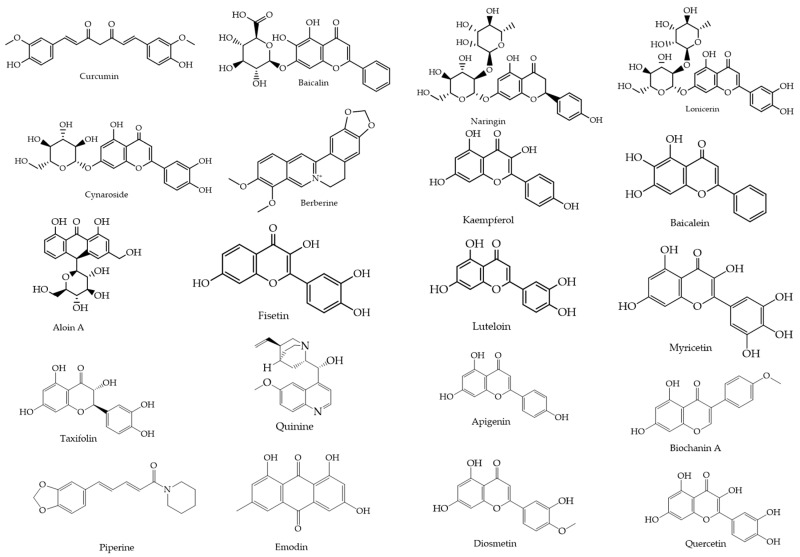
Chemical structures of the 20 selected flavonoids and phytocompounds evaluated.

**Figure 3 ijms-26-10541-f003:**
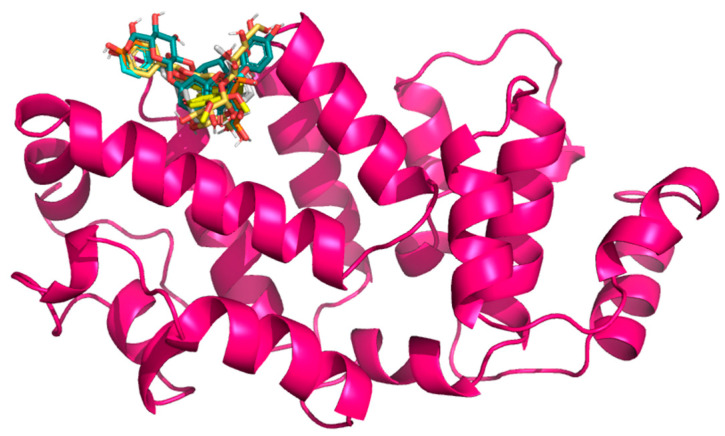
Three-dimensional structure of the OROV Gc glycoprotein head domain (PDB ID: 6H3X), with docked flavonoid compounds positioned in the predicted binding pocket.

**Figure 4 ijms-26-10541-f004:**
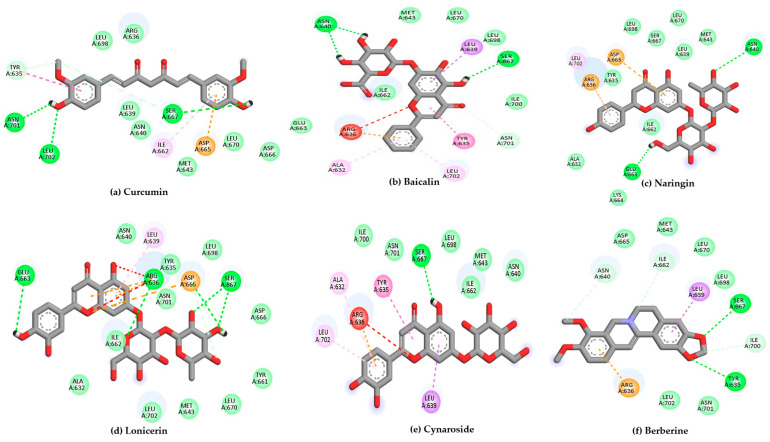
2D interaction diagrams of the top six ligand and positive control–Gc glycoprotein head domain complexes, selected for molecular dynamics simulations. * Positive control compound.

**Figure 5 ijms-26-10541-f005:**
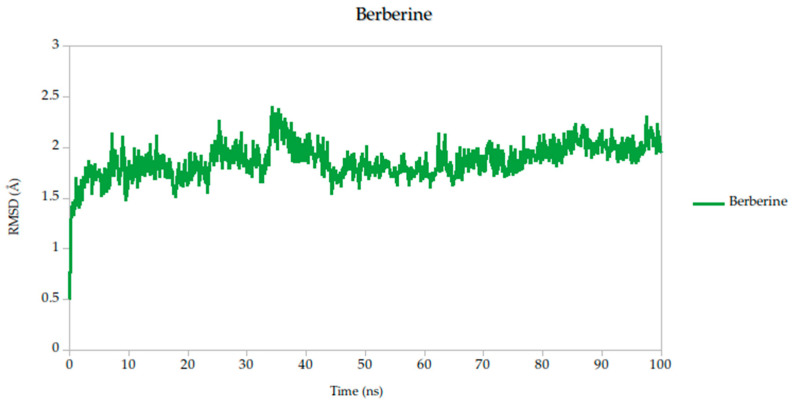
Backbone RMSD trajectories of Gc–Ligand complexes during 100 ns molecular dynamics simulations.

**Figure 6 ijms-26-10541-f006:**
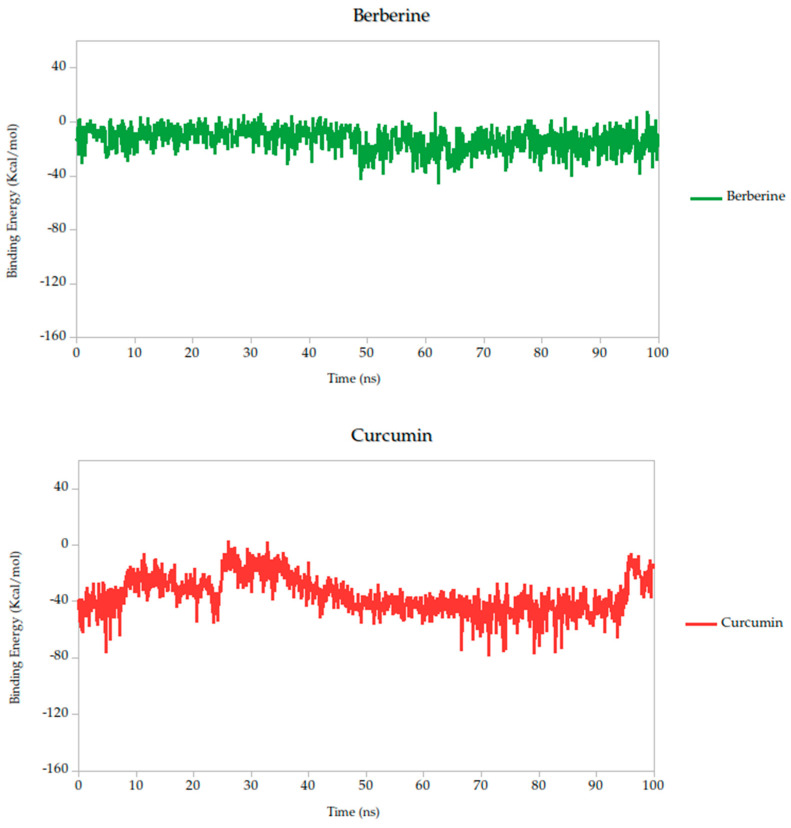
Binding energy dynamics of Gc–ligand complexes throughout the 100 ns simulation.

**Figure 7 ijms-26-10541-f007:**
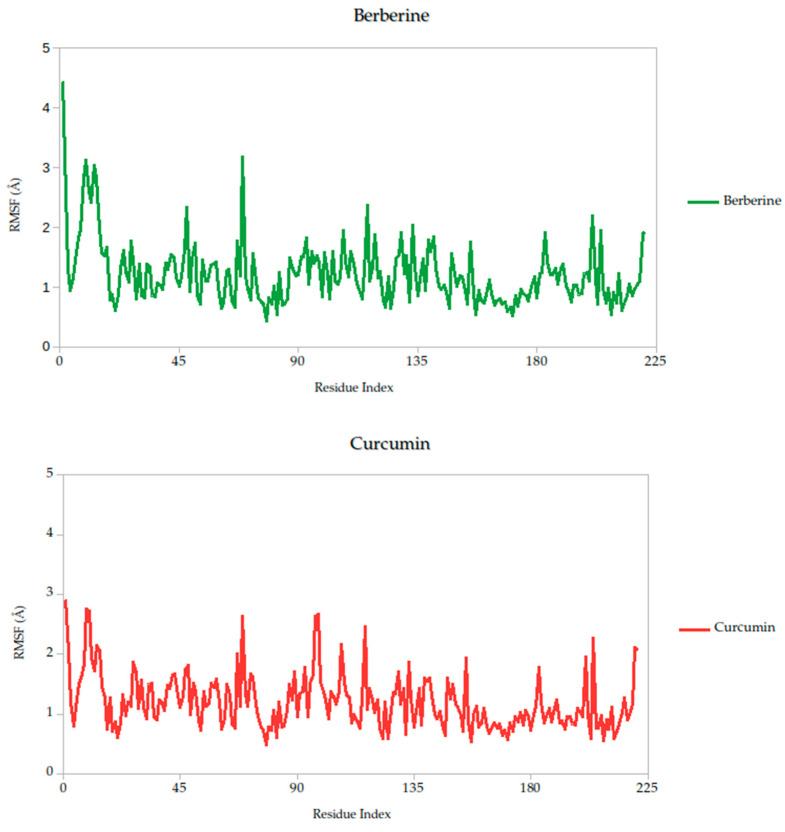
Per-residue RMSF profiles of Gc–Ligand complexes over 100 ns molecular dynamics simulations.

**Figure 8 ijms-26-10541-f008:**
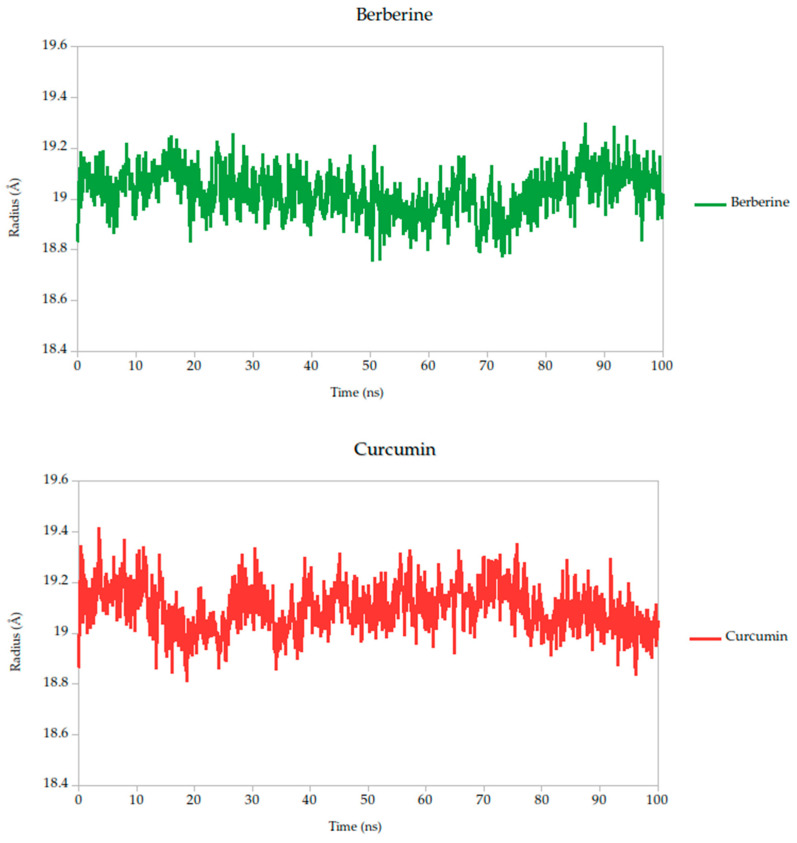
Radius of gyration (Rg) of Gc–Ligand complexes during 100 ns molecular dynamics simulations.

**Figure 9 ijms-26-10541-f009:**
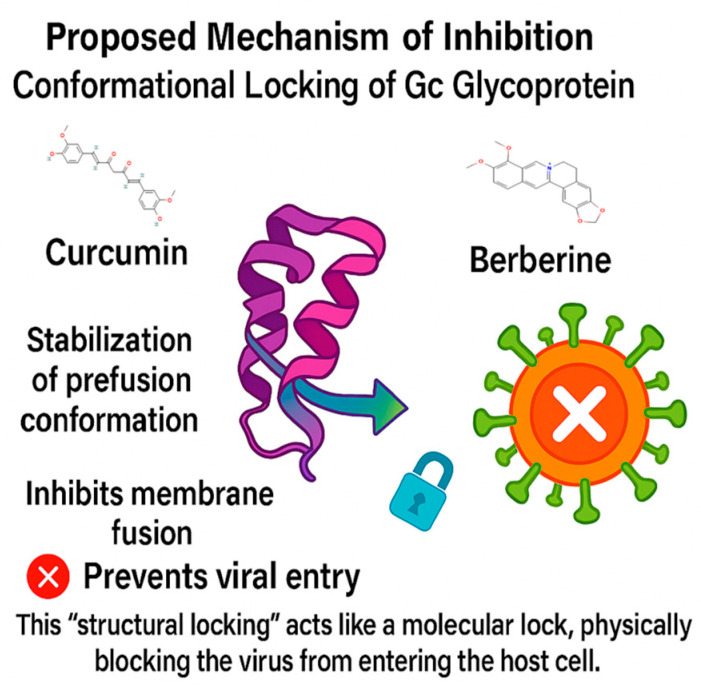
Curcumin and berberine inhibit OROV entry by stabilizing the Gc glycoprotein in its prefusion conformation, acting as a molecular lock that blocks membrane fusion.

**Table 1 ijms-26-10541-t001:** Docking results of 20 selected molecules against the OROV Gc glycoprotein (PDB ID: 6H3X).

Target	Ligands	Binding Energy (kcal/mol)	Hydrogen Bonds/Carbon Hydrogen Bonds	Other Interactions
(6H3X)	Curcumin	−7.9	Tyr635 Arg636, Asn640, Met643, Asp666, Leu670, Leu698, Asn701, Leu702	Ile662, Asp665
	Baicalin	−7.4	Asn640, Met643, Ile662, Glu663, Ser667, Leu670, Leu698, Ile700, Asn701	Ala632, Tyr635, Arg636, Leu639, Leu702
	Naringin	−7.3	Ala632, Tyr635, Leu639, Asn640, Met643, Ile662, Glu663, Lys664, Ser667, Leu670, Leu698	Arg636, Asp665, Leu702
	Berberine	−7.3	Tyr635, Asn640, Met643, Ile662, Asp665, Ser667, Leu670, Leu698, Ile700, Asn701, Leu702	Arg636, Leu639
	Lonicerin	−7.1	Arg636, Asn640, Ile662, Asp665, Leu639, Leu702	Ala632, Tyr635, Met643, Glu663, Ser667, Leu698
	Cynaroside	−7.1	Asn640, Ile662, Ser667	Ala632, Tyr635, Met643, Lys647, Glu663, Leu698, Ile700, Asn701
	Kaempferol	−6.9	Tyr635, Leu639, Asn640, Asp665, Ser667, Ile700	Met643, Tyr661, Asn701
	Baicalein	−6.9	Tyr635, Asn640, Asp665, Asp666, Ser667, Leu670, Asn701	Leu639, Met643, Ile662, Leu702
	Aloin A	−6.9	Tyr635, Asn640, Met643, Ile662, Glu663, Lys664, Asp665, Asp666, Ser667, Leu670, Leu698	Arg636, Leu639
	Fisetin	−6.8	Tyr635, Asn640, Ser667	Leu639, Met643, Ile662, Asp665, Leu670, Leu698, Asn701, Leu702
	Luteolin	−6.8	Tyr635, Asn640, Ser667	Leu639, Met643, Ile662, Leu698, Asn701, Leu702
	Myricetin	−6.8	Tyr635, Arg636, Ile662, Asp665, Ser667	Leu639, Asn640, Met643, Ile700, Asn701
	Quercetin *	−6.8	Tyr635, Ile662, Asp665, Ser667	Leu639, Asn640, Met643, Tyr661, Ile700, Asn701, Leu702
	Taxifolin	−6.8	Asn640, Ile662, Ser667, Leu702	Tyr635, Leu639, Met643, Asp665, Ile700, Asn701
	Quinine	−6.7	Tyr635, Asn640, Asp665, Ser667, Leu698, Asn701, Leu702	Arg636, Leu639, Met643, Ile662
	Apigenin	−6.7	Tyr635, Ser667	Leu639, Asn640, Met643, Leu670, Leu698, Asn701, Leu702
	Biochanin A	−6.6	Tyr635, Met643, Ile662, Ser667, Leu670, Leu698, Ile700, Asn701, Leu702	Ala632, Arg636, Leu639
	Emodin	−6.6	Tyr635, Arg636, Asn640, Ile662, Ser667, Leu670, Leu702	Leu639, Met643, Asp665
	Diosmetin	−6.6	Tyr635, Asn640, Ser667	Leu639, Met643, Lys647, Ile662, Leu698, Asn701, Leu702
	Piperine	−6.3	Tyr635, Leu639, Asp665, Ser667, Leu670, Leu698	Ala632, Arg636, Ile662, Leu702

* Positive control.

**Table 2 ijms-26-10541-t002:** Pharmacokinetic properties of the selected compounds: molecular dynamics.

Properties		Curcumin	Baicalin	Naringin	Lonicerin	Cynaroside	Berberine	Quercetin
Physico-chemical Properties	MW (g/mol)	368.38	446.36	580.53	594.52	448.38	336.36	302.24
	Num. Heavy atoms	27	32	41	42	32	25	22
	Num. Arom. Heavy atoms	12	16	12	16	16	16	16
	Num. Rotatable atoms	8	4	6	6	4	2	1
	Num. H-bond acceptors	6	11	14	15	11	4	7
	Num. H-bond donnors	2	6	8	9	7	0	5
Lipophilicity	Consensus Log P	3.03	0.25	−0.87	−1.03	0.16	2.53	1.23
Water solubility	Log S (ESOL)	−3.94	−3.41	−2.98	−3.64	−3.65	−4.55	−3.16
Pharmacokinetics	GI absorption	High	Low	Low	Low	Low	High	High
	BBB permeant	No	No	No	No	No	Yes	No
Druglikeness	Lipinski. Violation	0	2	3	3	2	0	0
Medicinal Chemistry	Synth. accessibility	2.97	5.09	6.16	6.37	5.17	3.14	3.23
Toxicity	Hepatotoxicity (probability)	Inactive (0.61)	Inactive (0.75)	Inactive (0.81)	Inactive (0.81)	Inactive (0.82)	Inactive (0.82)	Inactive (0.69)
	Neurotoxicity (probability)	Inactive (0.81)	Inactive (0.9)	Inactive (0.88)	Inactive (0.88)	Inactive (0.88)	Active (0.55)	Inactive (0.89)
	Nephrotoxicity (probability)	Active (0.59)	Active (0.68)	Active (0.77)	Active (0.77)	Active (0.76)	Inactive (0.5)	Active (0.62)
	Mutagenicity (probability)	Inactive (0.88)	Inactive (0.68)	Inactive (0.73)	Inactive (0.73)	Inactive (0.76)	Active (0.62)	Active (0.51)
	Cytotoxcity (probability)	Inactive (0.88)	Inactive (0.91)	Inactive (0.66)	Inactive (0.66)	Inactive (0.69)	Active (0.96)	Inactive (0.99)
	Predicted LD50 (mg/kg)	2000	5000	5000	5000	5000	200	159
	Tox-Level	4	5	5	5	5	3	3

**Table 3 ijms-26-10541-t003:** Binding energy and RMSD parameters for Gc–ligand complexes over 100 ns MD simulations.

	Curcumin	Baicalin	Naringin	Lonicerin	Cynaroside	Berberine	Quercetin
Binding Energy average (kcal/mol)	−38.14	−62.84	−58.01	−92.91	−50.81	−13.10	−39.79
σ Binding Energy (kcal/mol)	14.25	15.61	15.75	14.10	17.49	8.89	11.51
RMSD min (Å)	0.52	0.52	0.48	0.50	0.48	0.50	0.46
RMSD max (Å)	2.48	2.44	2.36	2.55	2.29	2.39	2.35
RMSD mean (Å)	1.82	1.79	1.78	1.87	1.82	1.86	1.91
σ RMSD	0.16	0.16	0.15	0.22	0.14	0.17	0.13

## Data Availability

The original contributions presented in this study are included in the article. Further inquiries can be directed to the corresponding authors.

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
