# Peer review of "Structure-Based Identification of Natural Inhibitors Targeting the Gc Glycoprotein of Oropouche Virus: An In Silico Approach"

_ijms, 2025, doi:10.3390/ijms262110541_

Round 1
Reviewer 1 Report
Comments and Suggestions for Authors
The manuscript presents a valuable in silico investigation of natural antiviral compounds targeting the Gc glycoprotein of the Oropouche virus (OROV), which the authors indicated as the first study to explore this protein using a structure-based computational approach. The authors apply a well-integrated computational workflow, including molecular docking, MD simulations, MM-PBSA estimations, and ADMET profiling, to identify promising plant-derived scaffolds, notably curcumin and berberine, as potential Gc inhibitors. The manuscript is methodologically adequate and addresses significant research information about therapeutic strategies for OROV.
The manuscript has the potential to be a valuable contribution for researchers interested in Gc inhibitors. However, to achieve this, the authors should reconsider the organization and presentation of the results to ensure the information is conveyed in a clear, coherent, scientifically accurate, and well-structured manner.
Given the significant potential impact of this type of study, the recommendation is rejection in its current form. However, it is strongly encouraged that the authors revise the manuscript thoroughly, addressing the comments and suggestions provided, and resubmit it for reconsideration.
Suggestions:
- Figure 1 needs to be redrawn because the compound structures are unclear, flattened, and of low resolution.
- The manuscript states that luteolin was selected as the baseline scaffold for similarity searches due to its well-documented antiviral activity and favorable structural properties (Section 2: Results, 2.1 and 2.1.1). However, in the docking and MD studies (Section 2.3), quercetin is used as the positive control, despite the authors also describing luteolin as a promising docking scaffold. This raises the question: Why was quercetin chosen as the reference compound instead of luteolin, which guided the initial compound selection? The authors should clarify this inconsistency and provide a rationale for using quercetin as the docking benchmark while using the library on luteolin. As listed in Table 1, both present the same binding energy, only with different interactions.
- The authors present essentially the same content regarding similarity-based compound retrieval using PubChem in two separate sections: Section 2.1.2: Similarity-Based Compound Retrieval Using PubChem: 2D and 3D Approaches (Results), and Section 3.1.1: Similarity-Based Compound Retrieval Using PubChem: 2D and 3D Approaches (Materials and Methods). This repetition introduces unnecessary redundancy. The authors should consolidate this information and ensure it is presented in the appropriate section, while only summarizing key outcomes in the Results.
- Lines 174-177, Section 2.3. Docking results, first paragraph: correct the Figure number, which is 3, not 2.
- Figure 2, in its current form, does not add meaningful value to the manuscript. Its relevance could be significantly improved by highlighting the active site and labeling the key interacting amino acid residues, which would help contextualize the docking and dynamics results.
- Figure 3 should be redone, keeping only the 2D interaction diagrams. The 3D representations, as currently presented, are not relevant — they lack clarity, the interactions are not visible, and in most cases, the compound is obscured by the ribbon structure.
- For consistency throughout the text, decide whether quercetin should be written with an uppercase or lowercase.
- It is unclear how the authors determined that Tyr570, Arg604, Ala605, Ser607, and Asn608 are key residues. They also affirm that Tyr 675 is a key residue. The manuscript does not provide literature-based justification for highlighting these residues as important. Moreover, the positive control quercetin (as the authors indicated) does not interact with any of them, which further raises questions about their relevance. There is also an inconsistency regarding the identification of key interaction residues. On page 11, during the MD analysis, the authors state that Tyr635, Met643, Asp665, and Leu702 form part of a conserved and potentially druggable pocket. This discrepancy needs to be addressed - the manuscript should clarify which residues are truly functionally relevant and explain how their importance was determined (e.g., through sequence conservation, structural role, or prior studies, or any other information that supports these affirmations). The text must be rewritten accordingly.
- Lines 281 -282: The authors need to verify if this is Figure 3? ”…….positions identified as part of a conserved and potentially 281 druggable pocket in the viral fusion protein (see Figure 3)”.
- The rationale for selecting only Curcumin, Baicalin, Naringin, Lonicerin, Cynaroside, and Berberine for ADMET evaluation is not clearly explained. Several other compounds — such as Biochanin A, Luteolin, and Cyanidin - show docking scores equal to or higher than –6.8 kcal/mol, yet were not included in this analysis.
- The authors should clarify the selection criteria used for ADMET profiling and justify the exclusion of other candidates with comparable or better binding affinities. Furthermore, the authors stated that these compounds are subject to MD investigations and in vitro validation.
- The toxicity of berberine is a concern, yet it is still promoted as a lead scaffold. Due the authors think that is appropriate to propose the berberine?
- Figure 5 is missing a caption. Why is quercetin missing in this picture? All the parameters must be compared with those of quercetin?
- Section 3.5. MD Simulations. Lines 443-445. The author indicated that they selected 6 compounds but listed only 5.
- The references need to be formatted consistently. As currently presented, they lack a unified template—some include full journal names while others use abbreviations; author name formats and punctuation also vary.
- Section: Introduction – Lines 54–55 are supported by 14 references, the following paragraph by 9 references, the sentence between lines 78–81 cites 16 references, and the sentence between lines 83-85 cites 20 references. In total, four paragraphs are backed by 59 references, which raises concerns about redundancy and over-citation. The authors should critically evaluate the relevance of each reference and retain only those that are essential and directly support the statements made.
Author Response
Referee #1:
The manuscript presents a valuable in silico investigation of natural antiviral compounds targeting the Gc glycoprotein of the Oropouche virus (OROV), which the authors indicated as the first study to explore this protein using a structure-based computational approach. The authors apply a well-integrated computational workflow, including molecular docking, MD simulations, MM-PBSA estimations, and ADMET profiling, to identify promising plant-derived scaffolds, notably curcumin and berberine, as potential Gc inhibitors. The manuscript is methodologically adequate and addresses significant research information about therapeutic strategies for OROV.
The manuscript has the potential to be a valuable contribution for researchers interested in Gc inhibitors. However, to achieve this, the authors should reconsider the organization and presentation of the results to ensure the information is conveyed in a clear, coherent, scientifically accurate, and well-structured manner.
Given the significant potential impact of this type of study, the recommendation is rejection in its current form. However, it is strongly encouraged that the authors revise the manuscript thoroughly, addressing the comments and suggestions provided, and resubmit it for reconsideration.
Answer: We would like to express our sincere thanks for your detailed and thoughtful review. We are truly grateful for the time and effort you dedicated to carefully reading our manuscript and for sharing such valuable and constructive feedback.
Your observations helped us recognize several aspects of our work that needed clearer explanation and better organization. We approached each of your suggestions with great care, and we have worked diligently to revise the manuscript accordingly. Beyond this specific revision, your comments have offered us important insights that we will continue to keep in mind, not only to improve this manuscript but also to strengthen the overall quality and rigor of our future research and writing.
We truly appreciate the opportunity to revise our work in light of your suggestions, and we are hopeful that the changes made reflect our sincere commitment to addressing your concerns thoughtfully and thoroughly.
Suggestions:
- Query: Figure 1 needs to be redrawn because the compound structures are unclear, flattened, and of low resolution.
Answer: We sincerely thank the Reviewer for this valuable suggestion. In response to your comment, we have redrawn the compound structures using ChemSketch and exported the image at high resolution (300 dpi) to ensure clarity and proper visualization of each molecule. The updated version now presents well-defined structures with improved contrast and visual consistency.
- Query: The manuscript states that luteolin was selected as the baseline scaffold for similarity searches due to its well-documented antiviral activity and favorable structural properties (Section 2: Results, 2.1 and 2.1.1). However, in the docking and MD studies (Section 2.3), quercetin is used as the positive control, despite the authors also describing luteolin as a promising docking scaffold. This raises the question: Why was quercetin chosen as the reference compound instead of luteolin, which guided the initial compound selection? The authors should clarify this inconsistency and provide a rationale for using quercetin as the docking benchmark while using the library on luteolin. As listed in Table 1, both present the same binding energy, only with different interactions.
Answer: We thank the reviewer for this important observation. Luteolin was indeed chosen as the baseline scaffold for the similarity-based compound retrieval (Section 2.1.1) due to its broad-spectrum antiviral properties, favorable pharmacokinetics, and well-described structural suitability for ligand-based design strategies (https://doi.org/10.1177/1934578X231171521; https://doi.org/10.1007/s11418-019-01287-7 ; https://doi.org/10.3390/ani13040761; https://doi.org/10.1186/s12917-024-04053-4 ; https://doi.org/10.1016/j.phymed.2023.155020; 10.3390/molecules14093486). This made it a logical starting point for the identification of structurally similar natural compounds.
However, for the docking studies and molecular dynamics simulations, quercetin was selected as the positive control because it has been previously reported in experimental studies to exhibit antiviral activity against Oropouche virus, providing a validated benchmark for comparison (https://doi.org/10.3390/biophysica3030032). While luteolin and quercetin showed similar docking scores (as reported in Table 1), the decision to use quercetin as a reference compound was based on its documented biological activity specific to OROV, which allows us to anchor the computational predictions with empirical antiviral data.
To avoid any confusion, we have clarified this distinction in the revised version of the manuscript, ensuring consistency between the scaffold-based library design and the reference used for performance comparison during docking and MD simulations.
- Query: The authors present essentially the same content regarding similarity-based compound retrieval using PubChem in two separate sections: Section 2.1.2: Similarity-Based Compound Retrieval Using PubChem: 2D and 3D Approaches (Results), and Section 3.1.1: Similarity-Based Compound Retrieval Using PubChem: 2D and 3D Approaches (Materials and Methods). This repetition introduces unnecessary redundancy. The authors should consolidate this information and ensure it is presented in the appropriate section, while only summarizing key outcomes in the Results.
Answer: We thank the reviewer for this observation. In response, we have revised the manuscript to eliminate redundancy. The detailed description of the 2D and 3D similarity search methodologies has been retained exclusively in the Materials and Methods section (3.1.1), where it is more appropriate. The corresponding paragraph in the Results section (2.1.2) was shortened significantly to briefly summarize the compound selection process and focus instead on the outcomes. We believe this modification enhances clarity and improves the overall structure of the manuscript.
- Query: Lines 174-177, Section 2.3. Docking results, first paragraph: correct the Figure number, which is 3, not 2.
Answer: Thank you very much for noticing this error. We have corrected the figure reference in Section 2.3, changing it from "Figure 2" to the correct "Figure 3" in the revised manuscript. We appreciate your careful reading and attention to detail.
- Query: Figure 2, in its current form, does not add meaningful value to the manuscript. Its relevance could be significantly improved by highlighting the active site and labeling the key interacting amino acid residues, which would help contextualize the docking and dynamics results.
Answer: Thank you sincerely for this valuable observation. In response, we have modified the title of Figure 2 to better reflect its role in the manuscript and have updated the figure to clearly display the binding pocket where the docking studies were conducted.
- Query: Figure 3 should be redone, keeping only the 2D interaction diagrams. The 3D representations, as currently presented, are not relevant — they lack clarity, the interactions are not visible, and in most cases, the compound is obscured by the ribbon structure.
Answer: We sincerely thank the Reviewer for this helpful recommendation. We fully agree that the 3D representations in the original Figure 3 lacked clarity and did not effectively illustrate the interactions. In response, we have removed the 3D views and now present only the 2D interaction diagrams, which clearly show the key residues involved in ligand binding. We believe this adjustment significantly improves the interpretability and visual quality of the figure.
- Query: For consistency throughout the text, decide whether quercetin should be written with an uppercase or lowercase.
Answer: Thank you for pointing this out. We have reviewed the manuscript carefully and corrected the inconsistencies in the capitalization of “quercetin,” ensuring that it is written consistently in lowercase throughout the text, as appropriate for compound names.
- Query: It is unclear how the authors determined that Tyr570, Arg604, Ala605, Ser607, and Asn608 are key residues. They also affirm that Tyr 675 is a key residue. The manuscript does not provide literature-based justification for highlighting these residues as important. Moreover, the positive control quercetin (as the authors indicated) does not interact with any of them, which further raises questions about their relevance. There is also an inconsistency regarding the identification of key interaction residues. On page 11, during the MD analysis, the authors state that Tyr635, Met643, Asp665, and Leu702 form part of a conserved and potentially druggable pocket. This discrepancy needs to be addressed - the manuscript should clarify which residues are truly functionally relevant and explain how their importance was determined (e.g., through sequence conservation, structural role, or prior studies, or any other information that supports these affirmations). The text must be rewritten accordingly.
Answer: We appreciate the reviewer’s detailed and constructive comment regarding the identification and relevance of the residues involved in ligand binding.
To clarify, the residues Tyr570, Arg604, Ala605, Ser607, and Asn608 were identified as interacting partners during the molecular docking of Curcumin, which showed the most strong interaction (dipole-dipole, hydrogen bond). These interactions were determined using both the docking pose visualization and interaction fingerprint analysis. We have revised the manuscript to specify that these residues are not claimed to be universally “key” residues for Gc functionality, but rather represent interaction points observed in the docking configuration of this particular ligand.
Separately, the residues Tyr635, Met643, Asp665, and Leu702 were recurrently involved in interactions across multiple top-ranked compounds, specifically those selected for further molecular dynamics (MD) simulations.
We have now clearly distinguished these two sets of residues in the revised text: one arising from the docking pose of curcumin, and the other from consistent patterns seen across compounds prioritized for MD simulation. Furthermore, we have eliminated ambiguous references to “key” residues unless supported by literature or structural conservation data, and instead describe the residues based on their context-specific interaction behavior.
These changes have been implemented throughout the manuscript to ensure scientific accuracy and clarity.
- Query: Lines 281 -282: The authors need to verify if this is Figure 3? ”…….positions identified as part of a conserved and potentially 281 druggable pocket in the viral fusion protein (see Figure 3)”.
Answer: Thank you for pointing this out. You are absolutely right, the reference should have been to Table 1, not Figure 3. We have corrected this in the revised version of the manuscript. We appreciate your careful reading, which helped us identify and amend this oversight.
- Query: The rationale for selecting only Curcumin, Baicalin, Naringin, Lonicerin, Cynaroside, and Berberine for ADMET evaluation is not clearly explained. Several other compounds — such as Biochanin A, Luteolin, and Cyanidin - show docking scores equal to or higher than –6.8 kcal/mol, yet were not included in this analysis.
Answer: We thank the reviewer for this thoughtful comment. The selection of Curcumin, Baicalin, Naringin, Lonicerin, Cynaroside, and Berberine for ADMET analysis was based primarily on their favorable binding energies (≤ –7.0 kcal/mol), along with a consistent pattern of molecular interactions across the docking analyses (https://doi.org/10.3390/molecules27249041). This multi-criteria approach aimed to balance docking performance with chemical diversity and biological plausibility, providing a rational basis for advancing the most promising candidates to pharmacokinetic evaluation.
- Query: The authors should clarify the selection criteria used for ADMET profiling and justify the exclusion of other candidates with comparable or better binding affinities. Furthermore, the authors stated that these compounds are subject to MD investigations and in vitro validation.
Answer: We thank the reviewer for the insightful comment. The six compounds selected for ADMET profiling (curcumin, baicalin, naringin, lonicerin, cynaroside, and berberine) were chosen based on their docking scores (≤ –7.0 kcal/mol) and their consistent interactions with relevant residues in the Gc glycoprotein binding site. Although other candidates showed comparable docking scores, they were not prioritized due to less frequent interaction patterns (https://doi.org/10.3390/molecules27249041). These selected compounds, along with quercetin as a positive control, were subsequently subjected to molecular dynamics simulations and are considered for future in vitro validation. The manuscript has been revised to reflect these selection criteria clearly.
- Query: The toxicity of berberine is a concern, yet it is still promoted as a lead scaffold. Due the authors think that is appropriate to propose the berberine?
Answer: Thank you very much for raising this important point. We fully agree that berberine’s potential toxicity is a relevant concern and should not be overlooked. Our intention was not to promote berberine as a definitive lead scaffold, but rather to include it among the evaluated candidates due to its favorable docking profile and its previously reported broad-spectrum antiviral activity. In fact, berberine has demonstrated inhibitory effects against several clinically relevant viruses, including SARS-CoV-2, dengue virus, influenza virus, enterovirus and herpesviruses, which highlights its pharmacological importance as a potential antiviral agent (10.3390/ijms23105690; https://doi.org/10.3390/molecules26185501; https://doi.org/10.3390/ijms22052368; https://doi.org/10.3390/v13061014; https://doi.org/10.3390/v13020282; https://doi.org/10.3390/molecules23082084)
However, in light of your observation, we have revised the discussion to better contextualize its inclusion, emphasizing that, despite its promising binding characteristics, its known cytotoxicity and narrow therapeutic index limit its direct therapeutic application. Nevertheless, we believe that the structural features contributing to its antiviral activity make berberine a valuable scaffold for rational drug design. Any future development involving berberine would require careful optimization to improve its safety profile.
- Query: Figure 5 is missing a caption. Why is quercetin missing in this picture? All the parameters must be compared with those of quercetin?
Answer: Thank you very much for your observation. We sincerely apologize for this error; the version of the figure initially uploaded was incomplete and did not include the caption or the data for quercetin. We fully agree that, as the reference compound, quercetin must be included for a meaningful comparison. The figure has now been corrected and properly updated in the revised manuscript. We will be especially careful to avoid such oversights in the future.
- Query: Section 5. MD Simulations. Lines 443-445. The author indicated that they selected 6 compounds but listed only 5.
Answer: Thank you very much for pointing this out. You are absolutely right. This was an oversight on our part. The text mistakenly mentions six compounds, when in fact only five were selected for MD simulations. We have corrected the sentence in the revised version to accurately reflect the number of compounds analyzed. We sincerely appreciate your attention to detail and will be more attentive to avoid such inconsistencies in the future.
- Query: As currently presented, they lack a unified template, some include full journal names while others use abbreviations; author name formats and punctuation also vary.
Answer: Thank you very much for your careful observation. We sincerely appreciate your attention to the formatting details. We have thoroughly reviewed and standardized all references to ensure consistency with the journal’s required style. This includes unifying the use of journal abbreviations, author name formats, and punctuation. We truly value your feedback and will be more meticulous in future submissions to prevent such inconsistencies.
- Query: Section: Introduction – Lines 54–55 are supported by 14 references, the following paragraph by 9 references, the sentence between lines 78–81 cites 16 references, and the sentence between lines 83-85 cites 20 references. In total, four paragraphs are backed by 59 references, which raises concerns about redundancy and over-citation. The authors should critically evaluate the relevance of each reference and retain only those that are essential and directly support the statements made.
Answer: Thank you very much for this thoughtful and important comment. We completely understand your concern regarding potential redundancy and over-citation. Upon your suggestion, we have carefully re-evaluated all the references cited in these paragraphs and have retained only those that are most relevant and directly support the statements made.
Reviewer 2 Report
Comments and Suggestions for Authors
This manuscript presents a scientifically sound in silico study identifying plant-derived compounds that may bind to the Oropouche virus (OROV) Gc glycoprotein and inhibit viral entry, offering potential as future therapeutics. With a few minor revisions to enhance mechanistic clarity, the paper would make a valuable and innovative contribution to the field of orthobunyavirus entry inhibition.
Comments added throughout the manuscript (uploaded document), including minimal updates to clarify content/readability and suggestions for additional details to bolster the conclusions.
Additionally, the resolution of the figures in the pdf make it difficult to see the compound structures clearly; suggest uploading higher quality images for the figures.

Author Response
This manuscript presents a scientifically sound in silico study identifying plant-derived compounds that may bind to the Oropouche virus (OROV) Gc glycoprotein and inhibit viral entry, offering potential as future therapeutics. With a few minor revisions to enhance mechanistic clarity, the paper would make a valuable and innovative contribution to the field of orthobunyavirus entry inhibition.
Comments added throughout the manuscript (uploaded document), including minimal updates to clarify content/readability and suggestions for additional details to bolster the conclusions.
Additionally, the resolution of the figures in the pdf make it difficult to see the compound structures clearly; suggest uploading higher quality images for the figures.
Answer: We sincerely thank the Reviewer for the encouraging and constructive feedback. We are pleased to know that our manuscript is considered scientifically sound and a potentially valuable contribution to the field of orthobunyavirus entry inhibition.
In response to your comments, we have carefully revised the manuscript to improve mechanistic clarity, refine the language for better readability, and incorporate the suggested details to further strengthen the conclusions. We also reviewed all figures and have replaced them with higher resolution images to enhance visual quality and clarity, particularly for compound structures.
We greatly appreciate the time and insight you invested in reviewing our work. Your thoughtful suggestions have helped us improve both the scientific content and presentation of the manuscript.
Suggestions:
- Query: Suggest updating with higher resolution images of the compounds (Figure 1 and Figure 3):
Answer: Thank you for your helpful suggestion. We agree that improving the visual clarity of Figure 1 and Figure 3 would enhance the overall quality and readability of the manuscript. In response, we have updated both figures with higher resolution images of the compounds to ensure better definition, improved visibility of structural details, and consistency across all representations.
- Query: How does your analysis of quercetin compare to the in silico modeling from the reference papers?
Answer: Thank you for this insightful question. Our analysis of quercetin aligns with previous in silico studies, including the work by Menezes et al. (https://doi.org/10.3390/biophysica3030032), which identified quercetin hydrate as a potential antiviral agent against Oropouche virus. In our study, quercetin demonstrated stable binding affinity and key interactions within the Gc binding site, supporting its use as a positive control and reinforcing its relevance in this context.
- Query: A conclusion figure (here or with conclusion) modeling your top compounds bound to Gc, with a mechanistic explanation for how this would prevent viral entry, would help drive home the conclusions.
Answer: Thank you for the helpful suggestion. We have now included a new conclusion figure (Scheme 1) illustrating the binding of our top compounds to the Gc glycoprotein, along with a schematic representation of the proposed mechanism by which this interaction may prevent viral entry, namely by stabilizing the pre-fusion conformation and blocking the conformational changes required for membrane fusion. We believe this addition strengthens the overall message of the study.
- Query: Compounds sometimes capitalized and sometimes lowercase - please make consistent throughout document (recommend lowercase).
Answer: Thank you for pointing this out. We have carefully reviewed the manuscript and standardized the capitalization of all compound names, using lowercase throughout the text in accordance with the journal's style and scientific writing conventions.
- Query: A more detailed summary of the class II fusion protein conformational change for orthobunyaviruses/OROV would be helpful either in the intro or here, to clarify why strong binding to Gc would be predicted to prevent viral entry.
Answer: We thank the reviewer for this insightful suggestion. We agree that providing a more detailed summary of the conformational changes undergone by class II fusion proteins, specifically in orthobunyaviruses such as Oropouche virus (OROV), strengthens the rationale for targeting the Gc glycoprotein. As described in recent structural studies (https://www.nature.com/articles/s42003-025-08040-9#citeas), OROV Gc remains in a metastable prefusion conformation on the virion surface, transitioning to a postfusion state upon endosomal acidification. This rearrangement involves insertion of the fusion loop into the host membrane and is essential for membrane fusion. High affinity binding of small molecules to regions critical for this transition, such as the fusion loop or trimerization interfaces, may stabilize the prefusion state and impede the conformational change required for viral entry. We have incorporated this explanation and cited the relevant literature to clarify the mechanistic basis for Gc-targeted inhibition.
- Query: A brief explanation on potential in vitro confirmatory experiments would add to the conclusion
Answer: We thank the reviewer for this valuable suggestion. In response, we have revised the conclusion section to briefly outline potential in vitro confirmatory experiments that could support our computational findings. Specifically, we now mention that future studies may include pseudotyped virus entry assays, cell–cell fusion assays, and plaque reduction neutralization tests using Gc-expressing systems. These experimental approaches would help validate the predicted inhibitory mechanisms and assess the compounds’ ability to block OROV entry.
- Query: Are there known structural similarities in Gc with other emerging orthobunyaviruses? Would be helpful to have a reference/specific information if that is predicted.
Answer: We thank the reviewer for this important observation. Notably, several orthobunyaviruses that infect humans, including Bunyamwera virus (BUNV), Germiston virus (GERV), Cache Valley virus (CVV), California encephalitis virus (CEV), Jamestown Canyon virus (JCV), Inkoo virus (INKV), and Ťahyňa virus (TAHV), encode Gc proteins that are predicted to adopt similar class II folds.
Recent computational and structural analyses have predicted that the Gc head domain in OROV retains a typical class II fold with functional homology to other orthobunyaviruses, supporting a shared mechanism of action (https://www.mdpi.com/1420-3049/30/3/503). Additionally, cryo-EM studies of SBV and AKAV glycoprotein complexes have revealed a striking structural conservation in their Gc subunits, further validating this premise (https://www.cell.com/cell-reports/fulltext/S2211-1247%2823%2900153-5) . Functional studies have also demonstrated that Gc mediates fusion in a pH-dependent manner across orthobunyaviruses, and neutralizing epitopes appear to cluster in homologous domains (https://doi.org/10.1186/s13567-015-0165-4; https://journals.plos.org/plospathogens/article?id=10.1371/journal.ppat.1003374).
Although these viruses are still considered neglected and remain understudied at the structural level, experimental data suggest that the head domain of the Gc glycoprotein is a major target of neutralizing antibodies. This highlights its immunological and therapeutic relevance and supports the rationale for selecting this domain as a target for small molecule docking in our study. We have incorporated this information and the cited literature into the revised manuscript to better contextualize the rationale for targeting Gc as a broadly relevant antiviral strategy.
Round 2
Reviewer 1 Report
Comments and Suggestions for Authors
The authors have addressed all of the reviewers’ comments thoroughly and implemented the necessary revisions accordingly. As a result, the revised manuscript now meets the publication standards and requirements of the IJMS.
Still, some minor requests need to be addressed:
- To enhance the clarity and overall understanding of the study, it is recommended that the authors include a workflow diagram that succinctly outlines the manuscript's main focus and methodology.
- Reference 82 needs to be checked – there is only a short text, no complete reference. In this case, authors should check whether they have 104 or 103 references. Also, reference numbers 80-85 are written 2 times at the beginning.
- The references still need to be checked, as there are still minor inconsistencies (e.g., references 77, 80, 81, 85, 86, 87, 88, 92, 100, 103, 104 compared to e.g., reference 99, where the title is formatted differently).
In this context, the manuscript is suitable for publication in IJMS pending the incorporation of these minor requests.
Author Response
We sincerely appreciate the reviewer’s positive evaluation of our revised manuscript and their helpful final observations. Please find below our point-by-point responses to the remaining comments:
Query 1: To enhance the clarity and overall understanding of the study, it is recommended that the authors include a workflow diagram that succinctly outlines the manuscript's main focus and methodology.
Response:
We thank the reviewer for this valuable suggestion. A workflow diagram has been created and included as Figure 1 in the revised manuscript (page 4). It summarizes the study's main objectives and methodology in a concise and visually accessible format. The figure has also been properly cited in the Introduction section.
Query 2: Reference 82 needs to be checked – there is only a short text, no complete reference. In this case, authors should check whether they have 104 or 103 references. Also, reference numbers 80–85 are written 2 times at the beginning.
Response:
We apologize for the oversight. The reference list has been carefully reviewed and corrected:
- Reference 82 has been completed with full citation details.
- Duplicate numbering of references 80–85 at the beginning has been resolved.
- The final reference count has been confirmed to be 104.
Query 3: The references still need to be checked, as there are still minor inconsistencies (e.g., references 77, 80, 81, 85, 86, 87, 88, 92, 100, 103, 104 compared to e.g., reference 99, where the title is formatted differently).
Response:
Thank you for highlighting this. We have now carefully reviewed and reformatted all references to ensure consistency in line with IJMS style guidelines. The previously mentioned references (77, 80, 81, 85–88, 92, 100, 103, 104) have been revised for uniformity in punctuation, capitalization, and title formatting.
We hope these changes address all remaining concerns. We are grateful for the constructive feedback and look forward to the final decision.